## Evidence synthesis 

psychology

obesity, behaviour change, cognitive training, policy

**Author for correspondence:**
Christopher D. Chambers
e-mail: chambersc1@cardiff.ac.uk

# Cognitive and environmental interventions to encourage healthy eating: evidence-based recommendations for public health policy

Lindsay A. Walker[1], Christopher D. Chambers[1], Harm Veling[2] and Natalia S. Lawrence[3]

[1]Cardiff University Brain Research Imaging Centre (CUBRIC), School of Psychology, Cardiff University, Cardiff, UK
[2]Behavioural Science Institute, Radboud University, Nijmegen, The Netherlands
[3]School of Psychology, University of Exeter, Exeter, UK

LAW, 0000-0002-1409-7101; CDC, 0000-0001-6058-4114;
HV, 0000-0002-5648-2980; NSL, 0000-0003-1969-6637

Policymakers are focused on reducing the public health burden of obesity. The UK average percentage of adults classified as obese is 26%, which is double that of the global average. Over a third of UK adults report using at least one weight management aid. Yet, many people still struggle to change their diet-related behaviour, despite having the awareness, intention and capability to do so. This 'intention–behaviour gap' may be because most existing dietary-choice interventions focus on individual decision-making, ignoring the effects of environmental cues on human behaviour. Behaviour change interventions that 'nudge' people into making healthier choices by modifying the food environment have been shown to be effective. However, this type of intervention is typically challenging for policymakers to implement for economic, ethical and public accessibility reasons. To overcome these concerns, policymakers should consider 'boosting' interventions. Boosting involves enhancing competences that help people make decisions consistent with their goals. Here, we outline cognitive training as a boosting intervention to tackle obesity. We synthesize the evidence for one type of cognitive training (go/no-go training) that may be effective at modifying food-related decisions and reducing body weight. We offer evidence-based recommendations for an obesity-focused Public Health Wales behaviour change programme.

# 1. Introduction

Obesity is a global public health threat. The World Health Organization estimated that in 2016, around 13% of the world's adult population were obese [1]. For the UK, the percentage of adults who are classified as obese was around double this: 26% in England, 2016 [2]; 27% in Northern Ireland, 2016/2017 [2]; 29% in Scotland, 2017 [3] and 22% in Wales, 2017/2018 (note: Wales is the only self-reported measure) [4].

The complexity of causal factors in obesity presents a challenge for intervention strategies [5]. To simplify for the purposes of brevity, human obesity probably reflects the interaction between multiple drivers that can be broadly categorized into two groupings: (i) biological and genetic factors, and (ii) nutrition and lifestyle/social factors [5–7] (see [8–10] for fuller discussion about the complex causes of obesity). For public health purposes, the latter is the most relevant. Within this, the focus of UK public health policy ranges from fiscal incentives and disincentives to the local built environment, almost all with the primary aim (or co-aim) of altering dietary choice [6]. The dietary choice itself is influenced by multilayered factors, including individual (e.g. genetics, age, gender, income), sociocultural (e.g. race/ethnicity, social and cultural norms), community environment (e.g. accessibility to and availability at supermarkets) and agricultural (e.g. food and beverage industry incentives) [6,11].

Most existing health interventions that target dietary choice focus on modifying individual autonomy in decision-making to persuade or dissuade certain choices [12,13]. For example, using the information to promote the avoidance of high sugar food via mass media campaigns or imparting skills to promote healthy cooking in weight-loss programmes. Yet, many people still struggle to change health-related behaviours despite having the awareness, intention and capability to make the changes [14,15]. For example, about 42% of adults worldwide report trying to lose weight [16], with 39% of UK adults reporting in 2016/2017 that they currently used at least one weight management aid [17]. This 'intention–behaviour gap' may be due to the fact that these 'top-down' approaches typically target conscious behaviour (i.e. individual autonomy in decision-making), which is in conflict with growing evidence that human behaviour is strongly directed by environmental stimuli, with the role of conscious intentions seeming to be limited [13,14,18]. For dietary choices, associative (Pavlovian and instrumental) learning mechanisms shape food preferences and food-related motivation for items high in fat, sugar or salt as these foods confer intrinsic rewards [19–22]. These automatic biases to attend to and approach rewarding foods can operate irrespective of an individual's dietary goal to eat healthily [23,24]—even when their intention to change eating is strongly desired [25]. In other words, we learn to approach and consume energy-dense foods associated with neural reward responses more strongly than low-calorie foods that do not elicit such a response. For these reasons, health-related behaviour change interventions typically have limited effects [13], and we argue that addressing the public health threat of obesity will require interventions that focus on targeting automatic cognitive processes.

Given the complexity around obesity and human behaviour change, it is important for any public health interventions to be based on robust evidence, particularly the evidence that considered what works in which circumstances (the 'difference-making' evidence; [5]). In this short paper, we explore evidence-based recommendations for public health interventions that target healthy-eating behaviour. The concept for this paper was initiated in response to the recent Welsh Government consultation on a draft strategy committed to helping people achieve and maintain a healthy weight, entitled *Healthy Weight: Healthy Wales* [26]. Specifically, our paper addresses part of the 'What Matters?' proposal under the Healthy People theme:

> Understanding what will encourage or prevent people from *adopting a healthier diet* or being physically active. We will work with Public Health Wales to design and deliver effective and high impact *behaviour change programmes* based on the evidence of what is effective for specific groups. [26, HP1] (author's own emphasis)

This paper focuses on 'adopting a healthier diet' as part of a behaviour change programme. First, we outline behaviour change approaches to altering food-related choices. Second, we explore the effects of cognitive training on changing people's 'automatic' behaviour towards food, focusing on response inhibition training as an example and including our own studies using online training and a fully developed training app. Finally, we outline evidence-based recommendations for policymakers, considering behaviour change programmes to tackle rising obesity levels, as with the Welsh Government's proposal.

# 2. Behaviour change approaches to altering food-related choices: the evidence

Evidence shows that decisions are often not deliberate, but instead are automatic and influenced by the environment [13,14]. Understanding how behaviour is mediated by the wider environment to stimulate unintentional responses, rather than focusing solely on a conscious decision, can inform and improve health-related behaviour interventions [12–14]. Applying behavioural insights to public health interventions is not an alternative to legislation—rather, it is both a toolkit and a wider approach to policymaking [14].

While there are many recommended behaviour change strategies for weight loss (e.g. [27]), one of the most well known is *nudging* [28]. Nudging involves making changes within the food environment that facilitate certain dietary-related choices over others. For example, increasing the visibility of healthier foods in a store can increase their consumption [29,30], increasing the proximity of food items so they are easier to reach increases their selection (e.g. [31]), and providing smaller plates may reduce the volume of food eaten (e.g. [32]). A recent meta-analysis of healthy-eating nudges suggests that behavioural strategies (e.g. altering the convenience or portion size of foods) have a threefold greater effect size than cognitive strategies (e.g. altering food visibility; [33]). The study estimates that this translates to a reduction of 13.4 teaspoons of sugar per day for behavioural-orientated interventions, compared to 4.0 less teaspoons of sugar per day for cognitively orientated interventions [33]. Additionally, interventions were found to be more effective when reducing unhealthy eating rather than promoting healthy eating [33]. Indeed, studies have found that promotions to reduce the consumption of relatively 'unhealthy' foods are effective *regardless* of a goal to restrict intake, whereas promotions to increase the consumption of relatively 'healthier' foods may be effective *only if* an individual has a dieting-related goal [25,34]. A substantial review by the McKinsey Global Institute indicated that supporting broad behavioural change to tackle obesity requires a '*systemic program of multiple interventions*' delivered by a wide range of sectors, with the more effective interventions relying less on individual decision-making and more on environmental changes (e.g. reducing portion sizes, changing food marketing practices; [35, p. 3]). A Public Health England review on reducing sugar intake reached similar conclusions, recommending a '*broad, structured programme of parallel measures*' that focused on environmental changes rather than individual willpower, such as restricting promotions and advertisements of energy-dense food [36, p. 7].

However, interventions that modify the food environment can be difficult for policymakers to implement. First, policymakers need to consider the economic concerns of food producers and sellers. Environmental interventions to reduce food consumption may impact profit margins (e.g. larger portions are more profitable than smaller portions as making more food costs less for companies through economy of scale), and it has been argued that policymakers need to also consider business profits to implement effective public health interventions [37]. Second, the interventions can raise ethical questions. Restricting food choice is likely to be the most effective environmental intervention, yet this limits people's freedom of choice, potentially altering human behaviour in a way that the person has not chosen. Third, the more effective environmental interventions may not be popular with voters, and as such may be hard for policymakers to legislate. For example, there was public outcry when a reduction in portion size by Toblerone was not coupled with a reduction in price [38]. Additionally, a survey of 693 adults of public support for sugar-reduction actions recommended by Public Health England (see [36]) revealed that a reduction in portion sizes was the least popular option [39]. Furthermore, education is rated a more acceptable food-related intervention by the public (in the UK and US) than nudge interventions or taxation [40]. Interestingly, communicating the evidence of policy effectiveness can increase public acceptability of interventions [40,41].

An alternative behaviour change approach that addresses these issues is *boosting* ([42]). Boosting involves enhancing the competences that ultimately help people make decisions consistent with their goals [43]. Note that instead of informing people what to eat and not to eat (a knowledge-based intervention), boosting focuses on the question of how to provide people with competences to change their actual behaviour in the direction they want. One way of doing this is by providing people with means to overcome the temptation to engage in unhealthy behaviours, so that healthy intentions can be successfully enacted. This approach may be easier for policymakers to implement for three key reasons. First, enhancing competences of the individual may be more straightforward than implementing environmental interventions in complex contexts such as the food environment. Second, people can opt to employ the competences to change their own behaviour in a manner of their

choosing, rather than having their options limited for them. Third, once an individual has acquired the competency, it can be applied to any situation with a food choice (e.g. restaurant, cafeteria, grocery store), rather than implementing environmental interventions in each context.

As outlined in the introduction, dietary preference for foods high in fat, sugar or salt (i.e. the food items that public health policies often aim to reduce the consumption of) is guided by learned processes that can act regardless of dietary intentions, potentially leading to unintentional or conflict-laden consumption of such food items. Perhaps due to such self-control 'lapses', people have suggested they would welcome interventions that help adherence to diet goals [33,44]. This represents an opportunity for boosting interventions. To enhance the competency of an individual to change their food-related behaviour, boosting interventions need to target the learned processes involved with food preferences, and give people access to such interventions so that they can use them when they want. To this end, cognitive training has recently been developed by researchers as a tool for individuals to modify their responses to food items (see [45] for a review). A range of cognitive training tasks have been developed that aim to modify attention, approach responses, reward value, working memory and inhibitory control towards energy-dense food [46–50] and alcohol [51]. Here, we focus on the effects of one type of cognitive training called go/no-go training (hereafter GNG), firstly because it has been found to be one of the most effective at modifying food-related choices, intake and reducing body weight in several studies (e.g. [46] for a meta-analysis), and secondly because it is the subject of our expertize.

# 3. Cognitive training to help modify food-related choices: a new cost-effective intervention for obesity?

GNG is a type of cognitive training intervention that involves modifying learned responses to certain food items such as those high in fat, sugar or salt (for a review and further information on the methodology see [52]). In brief, this is achieved by presenting the image of a food item alongside either a 'go' cue (that requires the participant to press a button in response to the image) or a 'no-go' cue (*not* pressing a button). By inhibiting the motor response (the 'no-go' cue), the training reduces the learned go/approach responses for the 'no-go' food item, and it reduces the liking for no-go items (e.g. [52]). Certain foods (e.g. unhealthy items) are consistently associated with 'no-go' cues, and other foods (e.g. healthy items) or non-food items are consistently associated with 'go' cues.

GNG training has been shown to be effective in reducing people's behaviour towards foods high in fat, sugar or salt, with meta-analyses, suggesting an effect size for basic studies in the medium range (Cohen's $d \approx 0.5$, see [46,53,54]). This, therefore, represents the potential for a new cost-effective intervention for obesity [45,52]. There are four aspects of food-related choices that the GNG training approach has been shown to effectively modify. First, food evaluation. Food items associated with 'no-go' cues typically receive reduced evaluations of how attractive or liked the item is, compared to both 'go' cue food items and food items not used in the training (e.g. [55,56]). For eating behaviour, this suggests that devaluation of no-go food items may be one of the critical mechanisms of action underlying GNG training, consequently reducing the appeal of otherwise tempting foods [52]. Second, food intake. When 'no-go' cues were associated with chocolate, participants consumed less chocolate following the training compared to individuals who did not receive this cue [57]. A similar reduction in food intake following GNG training has been shown for crisps [58,59]. Third, self-served portion size. Associating sweets with 'no-go' cues led to participants selecting less of the candies following training (up to 1 day earlier) compared to a control condition [60]. Fourth, food choice. Snack items associated with 'no-go' cues were chosen less following training compared to snack items associated with 'go' cues [61,62]. Chen *et al.* [61] also showed that the probability of choosing a healthful food (fruit or vegetable) over an unhealthy food (crisps and candy bars) could be increased by associating healthy foods with 'go' cues and unhealthy foods with 'no-go' cues.

GNG training has also facilitated participant weight loss. A study of 113 participants (mostly university students) revealed that individuals who received GNG training with 'no-go' cues associated with food and non-food items lost more weight than individuals receiving 'no-go' cues associated with non-food items only [50]. Another study of 83 participants from the community who were predominantly overweight or obese found that 'no-go' cues associated with energy-dense snack items produced significant weight loss for individuals at two-week and six-month follow-up [56], with a comparable effect size ($d = 0.48$) to earlier laboratory-based studies. Training also led to reduced energy intake (200 kcal less per day). In the only study of training effects in the general public to date,

GNG training was associated with reductions in self-reported weight (0.53–0.91 kg) and snacking on energy-dense foods (13–20%) at four to six weeks post training (based on preliminary results from more than 2000 people who provided complete data as part of the trial; [63]). This open, pragmatic trial delivers the GNG training via web-based training or a smartphone application, and has been used by over 100 000 people. However, as the trial has no control group for comparison, such preliminary findings await validation. This work is currently under way via a European Research Council grant held by members of our team [64,65].

Several of these studies indicate that the effects of GNG training are stronger for participants who are frequently dieting (e.g. [57,58,66]), or have a relatively high body mass index (e.g. [50]). Moreover, GNG training has been found to change food evaluation among people who are classed as morbidly obese, such that they rated 'no-go' foods as less attractive than 'go' foods [67]. Collectively, these findings suggest the potential for GNG training to be an effective addition to weight-loss programmes. That is, by offering GNG, people become competent to reduce their liking of tempting food items, which may help them to resist temptations. As yet, however, GNG training has had little 'real-world' testing. Additionally, logistical aspects, such as the optimum frequency, timing and duration of training and design elements, such as how to increase user engagement and enjoyment [68], require further research to understand how to maximize changes in eating behaviour. A recent study found that 'gamified' GNG training *reduced* the impact of the training on weight loss compared to the 'non-gamified' condition [69]. Although there is not enough evidence yet for GNG training to be the sole focus of a public health policy, the results presented here offer a promising option for cost-effective treatment of obesity and warrant serious consideration. Indeed, NHS England has recently launched a new digital platform, NHS Digital Apps Library, which helps people find health-related digital tools that have been tested and approved by the NHS based on '*evidence of effectiveness towards positive patient outcomes*' [70,71].

# 4. Evidence-based recommendations for policymakers

Based on the evidence presented in this article, we offer the following pragmatic recommendations for tackling obesity as part of government health policy, such as the Welsh Government consultation on the *Healthy Weight: Healthy Wales* strategy.

## 4.1. At the national level

— Policymakers should propose environmental *nudges* to promote healthy foods only when they are likely to be widely implemented and maintained. This would be when public health interventions are either also profitable for companies or enforceable with legislation. However, this may be challenging within the context of the food environment, as (i) interventions to reduce unhealthy eating are often not profitable, and (ii) legislation may reduce individual autonomy, which raises ethical issues as well as potentially being unpopular with voters.

— To overcome these concerns, policymakers should consider *boosting* interventions in addition to nudging strategies. Many people in the UK want to eat more healthily. In 2016/2017, 74% of obese adults and 53% of overweight adults stated that they were trying to lose weight, with 39% of all adults reporting that they currently used at least one weight management aid [17]. As such, boosting people's competences to achieve this goal, instead of implementing subtle environmental nudges that people cannot control, may be more valuable and cost-effective as well as more likely to be accepted by the wider public.

## 4.2. For the design of a public health behaviour change programme, such as proposed by Public Health Wales

— *Offer cognitive training as part of the behaviour change programme.* Such training fosters people's existing competencies or instils new ones, thus enabling them to modify their behaviour towards food in a way of their choosing to overcome unhelpful non-conscious processes. This helps people to overcome learned responses to food in a way that information and/or education cannot. Thus, cognitive training may help participants adhere to dietary goals across different contexts (e.g. eating at home,

eating out, work cafeteria). We would recommend consulting with cognitive training research experts to ensure effective training is deployed and that the effectiveness is appropriately monitored in a real-world setting.

— *Create a digital platform that disseminates evidence-based tools for managing weight.* Although education alone is not sufficient for changing people's food-related behaviour, it is a vital component of the broad programme required to tackle obesity effectively [35,36]. As such, offering people education on effective, evidence-based interventions that they can undertake themselves to change their own food micro-environment (e.g. at home and work cafeteria) would complement a wider strategy to tackle obesity.

Data accessibility. No data deposition is applicable for the paper.

Authors' contributions. L.A.W. drafted the article; C.D.C. revised it critically and provided a substantial contribution to design; H.V. revised it critically and provided a substantial contribution to conception; N.S.L. revised it critically and provided substantial contributions to conception. All authors gave final approval for publication.

Competing interests. C.D.C. is a member of the Royal Society Open Science editorial board but had no involvement in the peer-review process of this submission.

Funding. This work was supported by a grant from the European Research Council (Consolidator 647893; C.D.C.).

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
