## [Reviewer comments · Royal Society Open Science]

Review History

RSOS-190624.R0 (Original submission)

Review form: Reviewer 1

Is the manuscript scientifically sound in its present form?

Yes

Are the interpretations and conclusions justified by the results?

Yes

Is the language acceptable?

Yes

Is it clear how to access all supporting data?

Not Applicable

Do you have any ethical concerns with this paper?

No

Have you any concerns about statistical analyses in this paper?

No

Recommendation?

Accept with minor revision (please list in comments)

Comments to the Author(s)

Review of manuscript Cognitive training to encourage healthy eating: Evidence-based recommendations for a Public Health Wales behaviour change programme

MS ID: RSOS-190624

This is an interesting paper on a rapidly growing and public health relevant topic (the use of cognitive training to boost the ability to effortfully control diet). I agree with the authors' overall standpoint - public health interventions do need to address non-conscious processes or to equip people with the skills to overcome automatic, goal-incongruent behaviours.

General comments

1. Stating that people already have "the ability" to change diet-related behaviour (in amongst awareness and intentions, e.g. abstract, p2 of introduction, etc) seems odd given that the authors go on to describe a training intervention designed to enhance ability. The intention-behaviour gap is very much about the discrepancy between intentions and actions because of an inability to enact goal-directed actions. Rephrasing would be helpful.

2. Similarly, the authors describe boosting interventions as those which "help people make decisions consistent with their goals". This again, I feel should be rephrased. Many in the field of cognitive training would argue that training inhibitory control (as the described go/no-go training does) is much more about training the ability to suppress or inhibit behaviours that are not in line with goals but which are automatically cued by external factors, i.e. the training does not improve conscious, rational decision making per se. I would suggest rewording for clarity. For example, something along the lines of ".....boosting involves enhancing the cognitive skills that enable people to overcome the temptation to engage in unhealthy behaviours so that healthy intentions can be successfully enacted". If the authors agree, the phrasing would need amended throughout the manuscript.

3. The suggestion to introduce smaller sized crockery (while I agree, the available evidence suggests this would likely be helpful) comes out of the blue, seemingly unrelated to the main focus of boosting interventions. It would benefit from more of a lead in, or removal.

4. While I understand that the work was conducted in order to inform the decision making of Public Health Wales, is it necessary to retain a Wales-specific focus in the article? The issues outlined are of general relevance to public health. I would favour stating that the work was conducted in this context but amending the section from line 214 onwards to shift the focus from Wales to public health policy in general.

Minor comments

Line 99 - should that be 'reduce' rather than 'increase'?

Line 113 – presumably larger portions are cheaper through economy of scale, but this may not be immediately apparent to readers (why more food costs less for retailers). Worth making explicit for clarity.

Line 148 – Not sure that goal-setting interventions can be classed as non-conscious.

Line 155/156: Is it fair to badge go/nogo training as a ‘treatment for obesity’? It’s not a treatment as such.

Line 227 : Make clear that boosting interventions should be considered in addition to (as opposed to instead of) nudging strategies.

Line 237 – empowerment is a very conscious, deliberative process. I would be in favour of a shift to language that is consistently about the boosting of skills required to overcome unhelpful non-conscious processes.

Review form: Reviewer 2 (Michelle Lee)

Is the manuscript scientifically sound in its present form?

Yes

Are the interpretations and conclusions justified by the results?

Yes

Is the language acceptable?

Yes

Is it clear how to access all supporting data?

Not Applicable

Do you have any ethical concerns with this paper?

No

Have you any concerns about statistical analyses in this paper?

No

Recommendation?

Accept with minor revision (please list in comments)

Comments to the Author(s)

This evidence synthesis on inhibitory control training as an intervention for weight management and obesity prevention is directed to Public Health Wales in response to the Welsh Government consultation on its obesity strategy. Publication of evidence based recommendations is an excellent way to inform the debate and provide an important source of information for evidence based intervention decisions. This paper also provides an up-to-date and timely review of relevant literature for the academic community.

I have a couple of suggestions for the authors to consider:

1) I agree with the approach of ending the paper with some clear recommendations. However some of the recommendations are not really aligned with the evidence review in the manuscript

so the otherwise strong message about the value of 'boosting' interventions is overshadowed. For example a nudging recommendation is put forward based on only brief coverage of this literature which I read more as a scene setting for discussion of boosting strategies. Similarly there is a recommendation about using small plates and crockery more generally but this is not covered in the main body of the manuscript. Evidence is quite mixed on the plate size influence on intake - see Robinson et al. 2014 for a meta-analysis but my stronger point is that it seems off topic to include this. I would encourage the authors to focus more specifically on ICT/cognitive training so that it is centre stage as a recommendation.

2) It is a clear strength that the authors highlight the positive effects of ICT (specifically through GNG) on food selection, intake, and body weight. I was expecting to see some discussion of the likely effect size of ICT type interventions which would be especially relevant to policy makers. Can this be discussed based on recent meta-analyses? In addition should there be a discussion of the potential difference in effect sizes between lab-based studies and those carried out in real world settings if there is data to draw on here? Is this an area where caution is needed in terms of the likely benefits of ICT? Certainly Jones and Field (2018) suggest that ICT does not meaningfully reduce alcohol consumption in an ICT RCT.

3) The likely mechanism through which ICT works is briefly discussed and I would like to see this expanded with a more critical discussion. From the combined literature of ICT aimed at alcohol or eating there does not seem to be a consistent agreement that ICT works via stimulus devaluation (e.g. Di Lemma and Field, Houben et al. 2012, Lawrence et al. 2015 and the Jones et al. 2016 meta-analysis).

4) My final comment focuses on the broader introduction and context. At the beginning of the introduction the explanation of the population rise in obesity is put forward as an interaction between genes/physiology and life style/social choices and factors. It is important to recognise the complexity of obesity and the systems that underlie it and I think it is oversimplified here. I would not suggest a lengthy review of factors but an acknowledgement of the complexity and the challenge that presents for intervention strategies.

Review form: Reviewer 3

Is the manuscript scientifically sound in its present form?

Yes

Are the interpretations and conclusions justified by the results?

Yes

Is the language acceptable?

Yes

Is it clear how to access all supporting data?

Not Applicable

Do you have any ethical concerns with this paper?

No

Have you any concerns about statistical analyses in this paper?

No

Recommendation?

Accept with minor revision (please list in comments)

Comments to the Author(s)

The current paper describes recommendations for the Public Health Wales behaviour change programme. All in all, the current paper is very well written. However, please find some suggestions and feedback in the following.

The introduction contains a lot of relevant information, but it seems a bit disorganized to me and therefore it could be better structured. Also, in terms of strategies that are discussed here, the paper offers more than the cognitive training perspective. Therefore, I was wondering if this could be better reflected in the entire manuscript?

Please find my specific feedback in the following:

- Title: I find the title somewhat misleading as it suggests that the recommendations are solely about cognitive training. But the recommendations contain much more than this. It also contains adopting plate sizes, and providing a digital platform to provide evidence based tools to support weight loss. Also, in lines 76/77 when the authors describe their goals, there are more goals than "cognitive training". Maybe the authors could think about adapting it?

- In terms of using theory in interventions, here are some resources from a health psychology perspective that might be relevant to integrate and could add a slightly different perspective: Sniehotta et al. (2015).

<https://www.tandfonline.com/doi/full/10.1080/17437199.2015.1022902>

Prestwich et al., (2015):

<https://reader.elsevier.com/reader/sd/pii/S2352250X15001049?token=A313DE51DA66802606F76F33812CC8EAFD67B5ED771DCF65C2EA05C407CB905FCF16C27742D934D6ED0E6C238C91CFA4>

- lines 54 to 58: the authors make their point here quite clear that cognitive training is a good intervention strategy to target eating behaviour. But the same applies here, as for the title. Throughout the paper this seems that not only cognitive training is targeted and discussed here, but other strategies as well. The paper would be easier to follow, if that would be more transparent.

- lines 65 - 67: There might be more background information needed to understand the other paper/initiative that is mentioned here

- line 88: it seems to me that when only "nudging" as "the most well-known" behaviour change approach is mentioned, at lot of other approaches are falling short. NICE, for example, has a very well-developed guidance on individual behaviour change approaches

(<https://www.nice.org.uk/guidance/ph49>). I think that either, the topic needs to be a discussed on a broader scale, or, the authors could make more explicit why they are going to work with "nudging". It would also help to clarify much more throughout the text that it is about strategies that take place in an individual's environment. Because later, when it is about the cognitive training, then it is more about training the individual in making different choices.

- line 128: The authors describe boosting here and explain that it means to enhance an individuals competence to make more informed decisions. I was wondering what the difference between boosting and providing knowledge is? In terms of behaviour change interventions and the so-called intention behaviour gap, we know what knowledge is not a very good predictor for actual behaviour. Therefore, I was wondering how boosting as a strategy might minimise the intention-behaviour gap.

Decision letter (RSOS-190624.R0)

03-Jul-2019

Dear Dr Walker

On behalf of the Editors, I am pleased to inform you that your Manuscript RSOS-190624 entitled "Cognitive training to encourage healthy eating: Evidence-based recommendations for a Public Health Wales behaviour change programme" has been accepted for publication in Royal Society Open Science subject to minor revision in accordance with the referee suggestions. Please find the referees' comments at the end of this email.

The reviewers and handling editors have recommended publication, but also suggest some minor revisions to your manuscript. Therefore, I invite you to respond to the comments and revise your manuscript.

- Ethics statement

- Data accessibility

<http://datadryad.org/submit?journalID=RSOS&manu=RSOS-190624>

- Competing interests

- Authors' contributions

- Acknowledgements

- Funding statement

Because the schedule for publication is very tight, it is a condition of publication that you submit the revised version of your manuscript before 12-Jul-2019. Please note that the revision deadline will expire at 00.00am on this date. If you do not think you will be able to meet this date please let me know immediately.

- 1) A text file of the manuscript (tex, txt, rtf, docx or doc), references, tables (including captions) and figure captions. Do not upload a PDF as your "Main Document";
- 2) A separate electronic file of each figure (EPS or print-quality PDF preferred (either format should be produced directly from original creation package), or original software format);
- 3) Included a 100 word media summary of your paper when requested at submission. Please ensure you have entered correct contact details (email, institution and telephone) in your user account;
- 4) Included the raw data to support the claims made in your paper. You can either include your data as electronic supplementary material or upload to a repository and include the relevant doi

within your manuscript. Make sure it is clear in your data accessibility statement how the data can be accessed;

5) All supplementary materials accompanying an accepted article will be treated as in their final form. Note that the Royal Society will neither edit nor typeset supplementary material and it will be hosted as provided. Please ensure that the supplementary material includes the paper details where possible (authors, article title, journal name).

on behalf of Dr Christina Demski (Associate Editor) and Essi Viding (Subject Editor)
openscience@royalsociety.org

Associate Editor Comments to Author (Dr Christina Demski):

All three reviewers evaluated the manuscript positively. They also suggest some very useful revisions that would help better integrate ideas and widen the paper's relevance. Please address each point carefully if you choose to revise the manuscript.

Reviewer comments to Author:
Reviewer: 1

Comments to the Author(s)
Review of manuscript Cognitive training to encourage healthy eating: Evidence-based recommendations for a Public Health Wales behaviour change programme

MS ID: RSOS-190624

This is an interesting paper on a rapidly growing and public health relevant topic (the use of cognitive training to boost the ability to effortfully control diet). I agree with the authors' overall standpoint - public health interventions do need to address non-conscious processes or to equip people with the skills to overcome automatic, goal-incongruent behaviours.

General comments

1. Stating that people already have "the ability" to change diet-related behaviour (in amongst awareness and intentions, e.g. abstract, p2 of introduction, etc) seems odd given that the authors go on to describe a training intervention designed to enhance ability. The intention-behaviour gap is very much about the discrepancy between intentions and actions because of an inability to enact goal-directed actions. Rephrasing would be helpful.

2. Similarly, the authors describe boosting interventions as those which "help people make decisions consistent with their goals". This again, I feel should be rephrased. Many in the field of cognitive training would argue that training inhibitory control (as the described go/no-go training does) is much more about training the ability to suppress or inhibit behaviours that are not in line with goals but which are automatically cued by external factors, i.e. the training does not improve conscious, rational decision making per se. I would suggest rewording for clarity. For example, something along the lines of ".....boosting involves enhancing the cognitive skills that enable people to overcome the temptation to engage in unhealthy behaviours so that healthy intentions can be successfully enacted". If the authors agree, the phrasing would need amended throughout the manuscript.

3. The suggestion to introduce smaller sized crockery (while I agree, the available evidence suggests this would likely be helpful) comes out of the blue, seemingly unrelated to the main focus of boosting interventions. It would benefit from more of a lead in, or removal.

4. While I understand that the work was conducted in order to inform the decision making of Public Health Wales, is it necessary to retain a Wales-specific focus in the article? The issues outlined are of general relevance to public health. I would favour stating that the work was conducted in this context but amending the section from line 214 onwards to shift the focus from Wales to public health policy in general.

Minor comments

Line 99 - should that be 'reduce' rather than 'increase'?

Line 113 - presumably larger portions are cheaper through economy of scale, but this may not be immediately apparent to readers (why more food costs less for retailers). Worth making explicit for clarity.

Line 148 - Not sure that goal-setting interventions can be classed as non-conscious.

Line 155/156: Is it fair to badge go/nogo training as a 'treatment for obesity'? It's not a treatment as such.

Line 227 : Make clear that boosting interventions should be considered in addition to (as opposed to instead of) nudging strategies.

Line 237 – empowerment is a very conscious, deliberative process. I would be in favour of a shift to language that is consistently about the boosting of skills required to overcome unhelpful non-conscious processes.

Reviewer: 2

Comments to the Author(s)

This evidence synthesis on inhibitory control training as an intervention for weight management and obesity prevention is directed to Public Health Wales in response to the Welsh Government consultation on its obesity strategy. Publication of evidence based recommendations is an excellent way to inform the debate and provide an important source of information for evidence based intervention decisions. This paper also provides an up-to-date and timely review of relevant literature for the academic community.

I have a couple of suggestions for the authors to consider:

1) I agree with the approach of ending the paper with some clear recommendations. However some of the recommendations are not really aligned with the evidence review in the manuscript so the otherwise strong message about the value of 'boosting' interventions is overshadowed. For example a nudging recommendation is put forward based on only brief coverage of this literature which I read more as a scene setting for discussion of boosting strategies. Similarly there is a recommendation about using small plates and crockery more generally but this is not covered in the main body of the manuscript. Evidence is quite mixed on the plate size influence on intake - see Robinson et al. 2014 for a meta-analysis but my stronger point is that it seems off topic to include this. I would encourage the authors to focus more specifically on ICT/cognitive training so that it is centre stage as a recommendation.

2) It is a clear strength that the authors highlight the positive effects of ICT (specifically through GNG) on food selection, intake, and body weight. I was expecting to see some discussion of the likely effect size of ICT type interventions which would be especially relevant to policy makers. Can this be discussed based on recent meta-analyses? In addition should there be a discussion of the potential difference in effect sizes between lab-based studies and those carried out in real world settings if there is data to draw on here? Is this an area where caution is needed in terms of the likely benefits of ICT? Certainly Jones and Field (2018) suggest that ICT does not meaningfully reduce alcohol consumption in an ICT RCT.

3) The likely mechanism through which ICT works is briefly discussed and I would like to see this expanded with a more critical discussion. From the combined literature of ICT aimed at alcohol or eating there does not seem to be a consistent agreement that ICT works via stimulus devaluation (e.g. Di Lemma and Field, Houben et al. 2012, Lawrence et al. 2015 and the Jones et al. 2016 meta-analysis).

4) My final comment focuses on the broader introduction and context. At the beginning of the introduction the explanation of the population rise in obesity is put forward as an interaction between genes/physiology and life style/social choices and factors. It is important to recognise the complexity of obesity and the systems that underlie it and I think it is oversimplified here. I would not suggest a lengthy review of factors but an acknowledgement of the complexity and the challenge that presents for intervention strategies.

Reviewer: 3

Comments to the Author(s)

The current paper describes recommendations for the Public Health Wales behaviour change programme. All in all, the current paper is very well written. However, please find some suggestions and feedback in the following.

The introduction contains a lot of relevant information, but it seems a bit disorganized to me and therefore it could be better structured. Also, in terms of strategies that are discussed here, the paper offers more than the cognitive training perspective. Therefore, I was wondering if this could be better reflected in the entire manuscript?

Please find my specific feedback in the following:

- Title: I find the title somewhat misleading as it suggests that the recommendations are solely about cognitive training. But the recommendations contain much more than this. It also contains adopting plate sizes, and providing a digital platform to provide evidence based tools to support weight loss. Also, in lines 76/77 when the authors describe their goals, there are more goals than "cognitive training". Maybe the authors could think about adapting it?
- In terms of using theory in interventions, here are some resources from a health psychology perspective that might be relevant to integrate and could add a slightly different perspective: Sniehotta et al. (2015).
<https://www.tandfonline.com/doi/full/10.1080/17437199.2015.1022902>

Prestwich et al., (2015):

<https://reader.elsevier.com/reader/sd/pii/S2352250X15001049?token=A313DE51DA66802606F76F33812CC8EAFD67B5ED771DCF65C2EA05C407CB905FCF16C27742D934D6ED0E6C238C91CFA4>

- lines 54 to 58: the authors make their point here quite clear that cognitive training is a good intervention strategy to target eating behaviour. But the same applies here, as for the title. Throughout the paper this seems that not only cognitive training is targeted and discussed here, but other strategies as well. The paper would be easier to follow, if that would be more transparent.
- lines 65 - 67: There might be more background information needed to understand the other paper/initiative that is mentioned here
- line 88: it seems to me that when only "nudging" as "the most well-known" behaviour change approach is mentioned, a lot of other approaches are falling short. NICE, for example, has a very well-developed guidance on individual behaviour change approaches (<https://www.nice.org.uk/guidance/ph49>). I think that either, the topic needs to be discussed on a broader scale, or, the authors could make more explicit why they are going to work with "nudging". It would also help to clarify much more throughout the text that it is about strategies that take place in an individual's environment. Because later, when it is about the cognitive training, then it is more about training the individual in making different choices.
- line 128: The authors describe boosting here and explain that it means to enhance an individual's competence to make more informed decisions. I was wondering what the difference between boosting and providing knowledge is? In terms of behaviour change interventions and the so-called intention-behaviour gap, we know what knowledge is not a very good predictor for actual behaviour. Therefore, I was wondering how boosting as a strategy might minimise the intention-behaviour gap.

Author's Response to Decision Letter for (RSOS-190624.R0)

See Appendix A.

Decision letter (RSOS-190624.R1)

24-Sep-2019

Dear Professor Chambers,

I am pleased to inform you that your manuscript entitled "Cognitive and environmental interventions to encourage healthy eating: Evidence-based recommendations for public health policy" is now accepted for publication in Royal Society Open Science.

on behalf of Dr Christina Demski (Associate Editor) and Essi Viding (Subject Editor)
openscience@royalsociety.org

Appendix A

Response to reviewers

Please note: page numbers specified below refer to the revised manuscript with tracked changes.

For ease, the reviewer's comments have been numbered. Please find our responses to each below.

Reviewer 1

1.1 Stating that people already have “the ability” to change diet-related behaviour (in amongst awareness and intentions, e.g. abstract, p2 of introduction, etc) seems odd given that the authors go on to describe a training intervention designed to enhance ability. The intention-behaviour gap is very much about the discrepancy between intentions and actions because of an inability to enact goal-directed actions. Rephrasing would be helpful.

We thank the reviewer for noting this lack of clarity. Most people have the ability, or perhaps more accurately the capability, to make healthy eating choices but they often lack sufficient motivation or consideration. The COM-B model of behaviour change interventions (Michie et al., 2014) defines capability (psychological and physical) as an important determinant of behaviour. Psychological capability includes knowledge and psychological skills (planning, attention, stamina) and physical capability includes physical skills, strength and stamina that enable an individual to engage in a behaviour. Most individuals have the capabilities to eat more healthily but would benefit from boosting interventions to help with their automatic motivation to engage/enact these. To clarify this point, we have replaced instances of “ability” with “capability”.

1.2 Similarly, the authors describe boosting interventions as those which “help people make decisions consistent with their goals”. This again, I feel should be rephrased. Many in the field of cognitive training would argue that training inhibitory control (as the described go/no-go training does) is much more about training the ability to suppress or inhibit behaviours that are not in line with goals but which are automatically cued by external factors, i.e. the training does not improve conscious, rational decision making per se. I would suggest rewording for clarity. For example, something along the lines of “.....boosting involves enhancing the cognitive skills that enable people to overcome the temptation to engage in unhealthy behaviours so that healthy intentions can be successfully enacted”. If the authors agree, the phrasing would need amended throughout the manuscript.

We agree with the reviewer's suggestion to clarify the link between inhibitory control and meeting healthy eating goals. First, it is important to point out that the underlying mechanism of how GNG influences eating behaviour is not yet fully uncovered (for a discussion see e.g., reference 49 from the manuscript). Nonetheless, based on the available empirical evidence, it seems GNG influences eating behaviour by changing evaluations of food items such that no-go food items are devalued. From this perspective then, GNG does not train a cognitive skill (e.g., inhibitory control; which is also controversial; e.g., Inzlicht & Berkman, 2015), but it serves as a means for people to reduce their liking of tempting food items, which may in turn help them to act on healthy intentions. So by offering GNG, people become competent to change their liking of food items.

We have therefore have included the reviewer's suggested phrasing to modify as follows:

“Boosting involves enhancing the competences that ultimately help people make decisions consistent with their goals. ... One way of doing this is by providing people with means to overcome the temptation to engage in unhealthy behaviours, so that healthy intentions can be successfully enacted. ”

Later we state: “That is, by offering GNG, people become competent to reduce their liking of

tempting food items, which may help them to resist temptations.”
1.3 The suggestion to introduce smaller sized crockery (while I agree, the available evidence suggests this would likely be helpful) comes out of the blue, seemingly unrelated to the main focus of boosting interventions. It would benefit from more of a lead in, or removal.
We agree with the reviewer’s point that crockery should be removed and have done so. See also response to reviewers comment 2.1
1.4 While I understand that the work was conducted in order to inform the decision making of Public Health Wales, is it necessary to retain a Wales-specific focus in the article? The issues outlined are of general relevance to public health. I would favour stating that the work was conducted in this context but amending the section from line 214 onwards to shift the focus from Wales to public health policy in general.
We agree. We have rephrased the introduction and conclusion to have a broader focus, but have retained the mention of the Welsh Government as the impetus for this paper.
1.5 Line 99 – should that be ‘reduce’ rather than ‘increase’?
Thank you for spotting this error. This has now been corrected: “Indeed, studies have found that promotions to reduce consumption of relatively ‘unhealthy’ foods are effective regardless of a goal to restrict intake...”
1.6 Line 113 – presumably larger portions are cheaper through economy of scale, but this may not be immediately apparent to readers (why more food costs less for retailers). Worth making explicit for clarity.
We have added the suggested amendment.
1.7 Line 148 - Not sure that goal-setting interventions can be classed as non-conscious.
We agree and have removed goal-setting from this sentence.
1.8 Line 155/156: Is it fair to badge go/nogo training as a ‘treatment for obesity’? It’s not a treatment as such.
Thank you, we agree that this is potentially misleading and have replaced “treatment” with “intervention”.
1.9 Line 227 : Make clear that boosting interventions should be considered in addition to (as opposed to instead of) nudging strategies.
We have now inserted the suggested phrasing.
1.10 Line 237 – empowerment is a very conscious, deliberative process. I would be in favour of a shift to language that is consistently about the boosting of skills required to overcome unhelpful non-conscious processes.
We have replaced “empowering” with “enabling” and have inserted “.... Choosing to overcome unhelpful non-conscious processes”.

Reviewer 2

2.1 I agree with the approach of ending the paper with some clear recommendations. However some of the recommendations are not really aligned with the evidence review in the manuscript so the otherwise strong message about the value of 'boosting' interventions is overshadowed. For
--

example a nudging recommendation is put forward based on only brief coverage of this literature which I read more as a scene setting for discussion of boosting strategies. Similarly there is a recommendation about using small plates and crockery more generally but this is not covered in the main body of the manuscript. Evidence is quite mixed on the plate size influence on intake - see Robinson et al. 2014 for a meta-analysis but my stronger point is that it seems off topic to include this. I would encourage the authors to focus more specifically on ICT/cognitive training so that it is centre stage as a recommendation.

We agree with the reviewer's point that crockery should be removed and have done so. See also response to reviewer's comment 1.3

2.2 It is a clear strength that the authors highlight the positive effects of ICT (specifically through GNG) on food selection, intake, and body weight. I was expecting to see some discussion of the likely effect size of ICT type interventions which would be especially relevant to policy makers. Can this be discussed based on recent meta-analyses? In addition should there be a discussion of the potential difference in effect sizes between lab-based studies and those carried out in real world settings if there is data to draw on here? Is this an area where caution is needed in terms of the likely benefits of ICT? Certainly Jones and Field (2018) suggest that ICT does not meaningfully reduce alcohol consumption in an ICT RCT.

Thank you for this suggestion. We agree and have included information concerning the effect size estimates for basic studies and more recent real-world trials (pp8-9). We also note the need for caution in drawing conclusions about efficacy based, especially, on the preliminary nature of the real-world studies to date (p9).

2.3 The likely mechanism through which ICT works is briefly discussed and I would like to see this expanded with a more critical discussion. From the combined literature of ICT aimed at alcohol or eating there does not seem to be a consistent agreement that ICT works via stimulus devaluation (e.g. Di Lemma and Field, Houben et al. 2012, Lawrence et al. 2015 and the Jones et al. 2016 meta-analysis).

We agree that discussion of mechanisms is valuable. For the purposes of this evidence synthesis submission, we have deliberately kept such discussion to a minimum to maximise accessibility for a non-specialist (policy-focused) audience. However, in response to the reviewer's point we have revised the manuscript (p9) to note that, for eating behaviour specifically, devaluation appears to be at least one of the plausible mechanisms of action (we agree that there is uncertainty, especially viewed across the combined literature of eating and alcohol). See also our response to Reviewer 1.

There are at least 17 published experiments (in 7 papers; Chen et al., 2016; Chen et al., 2018a; Chen et al., 2018b; Lawrence et al., 2015; Serfas et al., 2017; Quandt et al., 2019; Veling et al., 2013) showing pre- to post changes in food value (devaluation) following food no-go training. The evidence may be more mixed for studies using implicit measures of devaluation or looking at the effects of training on alcohol devaluation, but it appears to be robust for studies using explicit measures of food devaluation, with 15 positive effects and no null effects published. While a file drawer of null results cannot be ruled out, this fits with wider evidence that no-go training results in stimulus devaluation but does not necessarily mean this is the one or only mediating mechanism for reduced food intake/choice.

2.4 My final comment focuses on the broader introduction and context. At the beginning of the introduction the explanation of the population rise in obesity is put forward as an interaction between genes/physiology and life style/social choices and factors. It is important to recognise the complexity of obesity and the systems that underlie it and I think it is oversimplified here. I

would not suggest a lengthy review of factors but an acknowledgement of the complexity and the challenge that presents for intervention strategies.

We agree with the reviewer that we have oversimplified the factors contributing to obesity for the purposes of brevity in the manuscript. As suggested, we now acknowledge the complexity and make clear that our statement is simplified, and point the reader in the direction of other papers that discuss the issue more fully.

Reviewer 3

3.1 Title: I find the title somewhat misleading as it suggests that the recommendations are solely about cognitive training. But the recommendations contain much more than this. It also contains adopting plate sizes, and providing a digital platform to provide evidence based tools to support weight loss. Also, in lines 76/77 when the authors describe their goals, there are more goals than “cognitive training”. Maybe the authors could think about adapting it?

In responses to comment 1.3 and 2.1 (see above), we have now removed the crockery recommendation. Even so, we agree with the reviewer that our recommendations extend beyond cognitive training alone, including nudges and a brief mention of education, therefore we have replaced “Cognitive training” in the title with “Cognitive and environmental interventions”

3.2 In terms of using theory in interventions, here are some resources from a health psychology perspective that might be relevant to integrate and could add a slightly different perspective: Sniehotta et al. (2015).

<https://www.tandfonline.com/doi/full/10.1080/17437199.2015.1022902>

Prestwich et al., (2015):

<https://reader.elsevier.com/reader/sd/pii/S2352250X15001049?token=A313DE51DA66802606F76F33812CC8EAFD67B5ED771DCF65C2EA05C407CB905FCF16C27742D934D6ED0E6C238C91CFA4>

Thank you for these references. We agree these are broadly of interest and would be highly relevant for a longer review article on behaviour change (and especially a critique of the role of theory, which we agree requires substantial development in understanding eating behaviour); however our article is not intended to focus on the role of theory but rather offer recommendations based primarily on empirical evidence for boosting interventions, and cognitive training in particular.

3.3 lines 54 to 58: the authors make their point here quite clear that cognitive training is a good intervention strategy to target eating behaviour. But the same applies here, as for the title. Throughout the paper this seems that not only cognitive training is targeted and discussed here, but other strategies as well. The paper would be easier to follow, if that would be more transparent.

We agree and have altered the title accordingly (see 3.1). Cognitive training is our major focus and the section noted by the reviewer (line 54–58 in the original submission) is intended to lay the groundwork for our more focused discussion of inhibitory control training.

3.4 lines 65 - 67: There might be more background information needed to understand the other paper/initiative that is mentioned here

We have rephrased the sentence to be clearer – see p4.

3.5 line 88: it seems to me that when only “nudging” as “the most well-known” behaviour change approach is mentioned, at lot of other approaches are falling short. NICE, for example, has a very well-developed guidance on individual behaviour change approaches

(<https://www.nice.org.uk/guidance/ph49>). I think that either, the topic needs to be a discussed on a broader scale, or, the authors could make more explicit why they are going to work with “nudging”. It would also help to clarify much more throughout the text that it is about strategies

that take place in an individual's environment. Because later, when it is about the cognitive training, then it is more about training the individual in making different choices.

We are grateful for this thoughtful comment. Concerning a specific point of emphasis, we note that we do not describe nudging as “the most well known” behaviour change intervention, but more cautiously as “one of the most well known”, which we feel is reasonable given the significant research literature and areas of policy devoted to nudges. We agree of course that the area of behaviour change extends well beyond nudging (as highlighted by NICE), and indeed beyond the scope of all interventions covered in this necessarily brief and focused submission. We have added a reference to the NICE guidelines (p5) to highlight this fact.

We agree that training cognitive control is individually focused, however there is also an essential environmental ingredient because training is intended to resist the environment by changing the liking of tempting foods, thus helping offset any detrimental effects of the environment rather than merely changing the individual's internal state or level of knowledge.

3.6 line 128: The authors describe boosting here and explain that it means to enhance an individuals competence to make more informed decisions. I was wondering what the difference between boosting and providing knowledge is? In terms of behaviour change interventions and the so-called intention behaviour gap, we know what knowledge is not a very good predictor for actual behaviour. Therefore, I was wondering how boosting as a strategy might minimise the intention-behaviour gap.

We thank the reviewer for noting this issue. To address this point, we have now expanded on the explanation of boosting to highlight the providing of competences in order to change behaviour versus merely providing knowledge.

We have now added: “Note that instead of informing people what to eat and not to eat (a knowledge-based intervention), boosting focusses on the question of how to provide people with competences to change their actual behaviour in the direction they want.” (p7)